# Regeneration of Vascular Endothelium in Different Large Vessels

**DOI:** 10.3390/ijms26020837

**Published:** 2025-01-20

**Authors:** Irina S. Sesorova, Eugeny V. Bedyaev, Pavel S. Vavilov, Sergei L. Levin, Alexander A. Mironov

**Affiliations:** 1Department of Anatomy, Ivanovo State Medical University, 153012 Ivanovo, Russia; irina-s3@yandex.ru (I.S.S.); akb37@mail.ru (E.V.B.); vavilov-007@mail.ru (P.S.V.); 2The EMC European Medical Centre, Central JSC RU, Spiridonyevsky Lane, 5, 123001 Moscow, Russia; doclevin2017@gmail.com; 3Department of Cell Biology, IFOM ETS—The AIRC Institute of Molecular Oncology, Via Adamello, 16, 20139 Milan, Italy

**Keywords:** endothelial cells, regeneration, artery, vein, lymphatic duct, cell cycle

## Abstract

The regeneration of endothelial cells (ECs) lining arteries, veins, and large lymphatic vessels plays an important role in vascular pathology. To understand the mechanisms of atherogenesis, it is important to determine what happens during endothelial regeneration. A comparison of these processes in the above-mentioned vessels reveals both similarities and some significant differences. Regeneration is carried out by moving intact ECs from the edges of the viable endothelial layer towards the centre of the EC damage zone. A sharp decrease in contact inhibition leads to the spreading of the edges of the ECs situated on the damage border. This stimulates the second row of ECs to enter the S-phase, then the G2 phase of cell cycle, and finally mitosis. In all three types of vessels studied, mitotically dividing ECs were found using correlation light and electron microscopy. These ECs have a body protruding into the lumen of the vessel, covered with micro-villi and other outgrowths. The level of EC rounding and protruding is highest in the arteries and least pronounced in the lymphatic vessels. The intercellular contacts of mitotically dividing cells become wider. The EC division leads to an increase in the density of ECs. ECs moving over the damaged area and partially outside the damaged area acquire a fusiform shape. *In the process of regeneration of arterial endothelium, the damaged ECs are removed. Then health ECs move to a surface devoid of endothelium, and detach spreading out, flattened platelets from the luminal surface of the vessel.* In the veins, ECs grow on the surface of platelets and microthrombi. In lymphatic vessels, ECs detach from the basement membrane slower than in the veins and arteries. There, the migrating ECs grow under fibrin fibres. After some time (usually after 30 days), the EC mosaic returns to normal in all three types of vessels.

## 1. Introduction

All vessels represent tubes, the walls of which most often consist of three parts: the inner (intima), the middle (media), and the outer (adventitia) wall. The inner lining of blood and lymph vessels is lined with endothelial cells (ECs). ECs are flat cells forming a continuous cell layer with pronounced polarity [1]. They are also characterised by the presence of special contact zones that contain tight junctions. ECs synthesise extracellular proteins, secrete them, and form a basement membrane (BM) containing mostly type IV collagen. The structure and composition of the membranes are determined by hemodynamic conditions. The integrity and correct three-dimensional organisation of the endothelial layer determine the permeability of the vascular wall and, ultimately, the normal course of all processes in the body [2].

ECs perform a variety of functions. They form a semi-permeable selective barrier for molecules, and they play a key role in the regulation of blood-clotting processes, as well as in inflammatory and immune reactions. The vascular endothelium secretes intercellular substance molecules (collagen types III, IV, and V, fibronectin, and laminin); adhesion molecules, mitogens, inhibitors and growth factors; and cytokines, which are capable of carrying out enzymatic lipolysis of lipoproteins, converting angiotensin-I, with the help of lytic enzymes, into small blood clots which are destroyed on the surface of ECs [2,3].

Regeneration, angiogenesis, and vasculogenesis represent other important functions of the vascular endothelium [4,5]. According to Ribatti et al. [6], it is useful to use the term “vascular niche”, which indicates the physical and biochemical microenvironment around a blood vessel, where ECs, pericytes, and smooth muscle cells (SMCs) organise to form blood vessels and release molecules involved in the recruitment of stem cells, endothelial progenitor cells, and mesenchymal stem cells. The impaired permeability and regeneration of the ECs in large arteries is one of the most important factors involved in atherogenesis and many other diseases. The damage of ECs in the cerebral artery is a key mechanism involved in the pathophysiology of various brain diseases, including stroke, atherosclerosis [7], diabetes mellitus [8], dyslipidaemia [9], hypertension [10], vasculitis [11], multiple sclerosis [12], and radiation necrosis [13].

Also, there are medical complications related to ECs [14,15]. The damage of ECs can lead to stroke [16]. The regeneration of ECs occurs after carotid endarterectomy [17,18], with vein transplantation [19,20,21,22,23,24,25,26,27,28]. Under all these conditions, proper endothelial repair is one of the main goals of treatment. The study of endothelial regeneration is important not only for understanding atherogenesis, but also for the prevention of venous damage after the use of catheters, after artery replacement with a vein, organ transplantation and the use of stents, and physical, chemical or mechanical damage to blood vessels that causes EC peeling [29,30,31,32,33,34].

Currently, interest in the study of the regeneration of ECs has been revived. For example, recently, Itoh et al. [15] investigated the EC regeneration in a 350 µm long segment of the middle cerebral artery, which was damaged as a result of a photochemical reaction. Transgenic mice with the green fluorescent protein Tie2 were used to identify ECs. Six hours after the endothelial injury, the ECs were separated from the lumen of the damaged artery, which was then covered with a layer of platelets. Within 24 h, the damaged artery began to recover from both edges, while the EC lengthened and migrated. The healing rate was higher at the proximal edge than at the distal one. Repeated endothelialisation with the EC proliferation peaked after 2–3 days and ended after 5 days. However, the study did not quote old pioneer articles. The authors mostly quoted review articles such as [35,36], although the most important information about the EC regeneration is contained in the original articles.

Significant progress in the creation of artificial blood vessels should be aimed at understanding the pathogenesis and developing treatment methods. Bioengineered arterial grafts with improved vascular functions have been developed. However, almost nothing is known about the formation of the endothelial coating of these grafts. Moreover, before considering this issue, it is necessary to know the cellular mechanism of endothelial regeneration.

This review provides a comprehensive overview of the knowledge about differences in the EC regeneration in different types of large natural vessels. Knowledge in this field has the potential to transform vascular medicine into more potent field and open up new possibilities for preclinical analysis. Our review is at least partially based on well-known old facts. However, it seems that most of them have been firmly forgotten, as these results are not cited. In modern reviews—for example, in the article by Ierka et al. [37]—the pioneering works by S. Schwartz, M. Reidy, and others (see below) are not cited and seem to be completely forgotten. Although all this was known a long time ago, these articles have not been cited until now. However, these results deserve to be cited. Therefore, this review is necessary, at least in order to make sure that the textbooks were written correctly and to remind us of who the pioneers were.

One of the aims of this review is to recall old and almost forgotten facts, compare EC regeneration in vessels of various origins, and increase scientific interest in studies involving tissues. We do not describe the mechanisms of restoration of EC monolayers in cell culture in this paper, because in vitro ECs behave like polarised epithelium, and the mechanisms are similar. The molecular mechanisms of EC regeneration in vitro are described. Many articles and reviews describe the regeneration of various ECs in vitro. For example, the equine aortic ECs exhibit an angiogenesis reaction in response to FGF2, but not to VEGF-A [34]. We have neither the place nor the desire to repeat these well-known facts here. Also, we have not analysed the mechanisms of the collective movement of polar cells after damage to the drainage layer in vitro (as there are many excellent reviews).

## 2. Regeneration of Arterial Endothelium

The detailed description of the cell secretory pathway in vascular ECs was previously presented [2]. It is important to note that in large arteries, BM has pores [2].

To initiate the process of endothelial regeneration, various deendothelialisation methods were used, which include the drying of the vessel lumen [38,39], the mechanical compression of the vessel, or the insertion of catheters into the lumen to destroy ECs [40,41,42,43,44,45,46,47,48], as well as hypotonic solutions or air enter the insulating area of blood vessels, the local freezing of blood vessel walls, and various chemicals and toxins, such as fatty acids [49,50] or nicotine [51].

Cotton et al. [4] were one of the pioneers in the study of regeneration of aortic ECs. Endothelial regeneration was studied mainly by groups led by S. Schwartz and M. Reidy [52,53,54,55,56,57,58,59,60,61], as well as by A. Mironov and M. Rekhter [62,63,64,65,66,67,68,69,70,71] more than 45 years ago. Schwartz’s group mainly focused on the mechanical damage of ECs. Our group used vessel freezing. This method was proposed by Malczak and Buck [72]. We improved this method. The freezing of vessels with the very cold profiled copper block causes the formation of ice crystals inside the ECs in a strictly defined area. These crystals cause the rupture of the plasma membrane and the death of the ECs. The freezing of the vessel wall does not damage the extracellular structures. In addition, this creates a very clear area of damage and a uniform front of the living ECs. Within 5 min after freezing and subsequent restoration of blood flow, the damaged ECs detached. BM becomes nude. This leads to the attachment of platelets and leukocytes to BM.

After 8–12 h, signs of the EC migration were detected in the marginal region of the intact endothelium. Migrating ECs form the leading edge, with lamellipodia in the anterior part. Lamellipodium detaches platelets from BM. The leading ECs have acquired a spindle shape, and their long axis is parallel to the axis of the vessel. By 13–24 h, ECs begin to synthesise DNA and divide [62,63,64,65,66,67,68,69,70,71]. In the dividing mitotic ECs, contact with their neighbours became wider [71]. This type of phenomenon is also observed in areas with turbulent blood flow in the human aorta [65].

However, in areas with laminar blood flow, the rate of the EC replication is quite low [72]. Endothelial regeneration occurs in accordance with a two-dimensional model of mesenchymal cell migration called “contact-stimulated migration” [73]. The first 4–7 resident EC lines present near the edges on both sides of the lesion synchronously entered the S-phase cell cycle. Their division greatly contributed to the restoration of the damaged area. By the end of the second day after the injury, a distinct hyper-plastic zone is formed, in which cells enter the S-phase of the cell cycle only after 20–28 h. The mitotic activity of cells reaches its maximum on the third day. The EC movement after this damage is independent of the EC division. Indeed, when the EC proliferation was suppressed by radiation, the compensatory proliferation of neighbouring ECs was observed [68]. This has also been shown in vitro [74].

The typical feature of damaged area after reendothelialisation is a change in the packaging of ECs. As a rule, the restored monolayer is characterised by an elongated shape of ECs and their increased density. Proliferating ECs have the shape of an elongated ellipsoid with a long axis parallel to the blood flow. In these cells, the number and size of mitochondria, the volume of the cytoplasmic reticulum, and the number of ribosomes increase. Hypertrophied nuclei are found in the cytoplasm.

After the monolayer is restored, the numerical density of ECs decreases due to the death and desquamation of some ECs and the spread of their neighbours. The process of subsequent differentiation of ECs in the reendothelialisation zone consists of a decrease in the specific volume of organelles of the biosynthetic and mitochondria, the formation of near-contact thickenings of microfilaments, and the formation of focal cell contacts with structures of the subendothelial layer [62,63,64,65,66,67,68,69,70,71].

An important aspect of endothelial regeneration in large arteries is the anisotropic nature of this process in different directions. In particular, the regeneration of the endothelium occurs at a much higher rate parallel to the direction of blood flow. This phenomenon can be explained by several factors, including: 1. The formative effect of blood flow on the regeneration process; 2. By modulating the substrate of the main tissue during regeneration; 3. The longitudinal orientation of the cytoskeletons of ECs; 4. The transverse orientation of cell division during regeneration. These factors, as well as others, may contribute to the observed anisotropy of endothelial regeneration in the context of large vessels [38]. When EC proliferation was suppressed by radiation, the compensatory proliferation of neighbouring ECs was observed [68].

Repeated injuries also lead to an increase in the degree of heterogeneity of ECs and the appearance of clusters of the multinuclear ECs. This phenotype was also observed in human aortae in areas with turbulent blood flow [65]. However, in areas with laminar blood flow, the overall basal rate of the EC replication is almost negligible [72]. Of interest is that, after the repeated freezing of the aorta, EC regeneration occurred faster than after mechanical damage of the aortic intima, since mechanical damage impairs BM, whereas BM was not damaged after freezing [38,60]. Also, ECs synthesise components of BM, making it thicker. ECs in the G2 phase show signs of secretion of BM components. They have ER exit sites. The thick BM had smaller pores and could better meet the needs of migrating ECs. This effect is observed in old animals [75] and in animals with arterial hypertension [38]. It is important to note that during initial phases of EC regeneration, smooth muscle cells (SMCs) are absent in intima, since SMCs in media were also killed by freezing [52,53,54,55,56,57,58,59,60,61,62,63,64,65,66,67,68,69,70,71].

There was no regenerative reaction in the medial layer of the cryo-damaged vessels. This layer consists only of elastic and collagen fibres. Only after the complete restoration of the EK monolayer do SMCs appear in the intima, which form thickenings. Migrating SMCs form spindle-shaped projections consisting of elongated SMCs. Their tip is directed towards the damaged areas already covered by the endothelium. The thickness of such SMC layers increases over time [38]. The scheme of arterial reendothelialisation and the restructuring of the SMC layers is shown in Figure 1G,H.

The molecular mechanisms involved in the regeneration of the aortic endothelium have been described in detail [34,73], especially in vitro. The proteins involved in EC are well described [34,75,76,77,78]. For example, the deletion of Atf3 reduced endothelial proliferation and disrupted regeneration. There were differences in the set of transcription factors that increased the activity of cyclins and CDK, as well as suppressed the transcription of p21 [77]. The collective migration of ECs is called ‘contact-stimulating migration’ [79]. The mechanism of lamellipodium formation is also well studied. They are not much different. Growth factors stimulate the migration of the capillary ECs, while the same factors do not affect the migration of the aortic ECs [80]. LDL can also enter the intima through open inter-endothelial contacts around mitotically dividing ECs [38].

Although the idea of circulating ECs is promising [54], more and more evidence indicates that circulating STEM cells do not directly contribute to endothelial regeneration and are not found in the newly formed endothelial zone. The existing consensus suggests that endothelial precursors exist [81]. However, the involvement of these cells in the regeneration of the aortic endothelium has not been proven [73]. When the resident ECS were damaged by radiation, bone marrow-derived cells were recruited and incorporated into the damaged vasculature, but these cells were unable to differentiate into ECs [34].

## 3. Regeneration of Venous Endothelium

The ECs of large veins have a typical structure, well-developed tight junctions, and a BM without pores. However, their BM is quite thin [40,82]; In the world, there are millions of cases of EC regeneration in veins every day. However, studies on this process are limited. There were no significant differences in the response of the venous and aortic ECs to damage caused by freezing. The reaction of inferior vena cava ECs to damage caused by freezing was studied. First, the deendothelialised surface is covered with flattened platelets, and then, three days after surgery, the endothelium is restored as a result of the migration and proliferation of ECs. ECs enter the S-phase (Figure 2A).

Migrating ECs do not remove platelets and fibrin fibres from the BM surface. Young ECs form a single layer of highly elongated ECs, the axis of which is parallel to the blood flow. An immature endothelium is characterised by an increased number of ECs. When crawling over fibrin deposits, ECs of the first two rows grow intensively and form numerous processes that cover fibrin [83]. After reendothelialisation, endothelial hyperplasia gradually decreases. It is accompanied by the appearance of giant multinuclear ECs. Seven days after the clamp damage, the defect with a diameter of 3 mm is completely covered by ECs [83,84,85,86]. The following reduction in ECs proceeds according to the type of cellular hyperplasia (an increase in the mitotic coefficient in ECs) and hypertrophy (an increase in nuclear and cellular sizes, the presence of ECs with two nuclei), which reflects some processes of intracellular renewal. The regeneration process has a late development and a long course, depending on the duration of pathological changes in the tissue [86].

When a section of the posterior vena cava of the rat was subjected to mechanical action, the reendothelialisation showed some peculiarities. After the release of the clamp, a deendothelialised surface is gradually formed within 4 h, containing wall microthrombi in places of deep intima injuries, which consist of a dense three-dimensional fibrin network with blood cells included in it. The deendothelialisation of the vein wall, associated with damage to the subendothelial layer, causes the formation of fibrin-rich wall microthrombi. As a result of the recovery of their surface, three-dimensional structures are formed in the lumen of the vessel’s endovasal growths [84,85]. The reendothelialisation of these cavities inside the thrombus leads to the formation of intramural channels that anastomose with each other and communicate with the lumen of the vein. Of interest is that, after the wall of the posterior vena cava underwent mechanical damage by applying a clamp, intramural angiogenesis was observed. 

During the interaction with separately located fibrin filaments branching out into the lumen of the vein or thicker bundles of fibrin, ECs cover them from all sides. The covering of fibrin bundles led to the formation of strands covered with endothelial monolayer in the lumen of the vessel—“endovasal structures”. In the process of the regeneration of ECs along the fibrin matrix of the parietal thrombus, which is complexly organised in space, various layers of ECs are formed on top of each other due to the growth of ECs at different levels of the fibrin matrix (Figure 2A–J).

In this case, BM is destroyed, and the endothelium migrates into the vessel wall. Endothelial cords are formed, which are channelled. At the same time, the material forming the basement membrane is deposited on the basal surface, and a capillary is formed in the vessel wall. Although, to be precise, it certainly cannot be called a true capillary. At the same time, intravascular growths are found in the lumen of the veins, which are a system of strands that are covered on all sides with endothelium, and spread from one wall of the vessel to another (Figure 2I,J).

The inner surface of the vein at the site of the clamp is represented by a variety of “endovasal cables”, as well as numerous niches and funnel-shaped depressions—the mouths of intramural channels. ECs located on “endovasal structures” retain signs of poorly differentiated ECs, whereas ECs of flat areas practically do not contain elements of a granular cytoplasmic network, which have more caveolae (Figure 2F–H,J).

The intramural channels really originate from numerous irregularly shaped mouths located on the reendothelialised surface of the vena cava. Then, in the form of anastomosing vessels between themselves, which is also seen on total preparations, the channels lie for a sufficiently long distance under the endothelium of the main vein. At the same time, they can anastomose with the perivascular vascular network, branch out, gradually decrease in diameter, and also blindly end in the thickness of the venous wall. Sometimes, intramural channels may have an irregular sinus-like shape. Intramural channels, with all other options, can again “return” to the vena cava.

By 14 days, ECs are constantly found at the entrances to the mouths of intramural channels, densely covered with microvilli, which is characteristic of dividing ECs, whereas in flat areas, almost all ECs have a flat surface. Mitoses in these areas are also visible on semi-thin sections.

A month later, the fibrin backbone of the growths is completely replaced by mature connective tissue, which contains coarse collagen fibres. At the same time, there is a decrease in the number and size of “endovasal structures”—there is a kind of “shrinkage” of them. On the contrary, the degree of vascularisation of the venous wall, reflecting the specific volume of intramural channels, increases somewhat, and their enlargement is observed with the appearance of sinus-like forms, an increase in the number of anastomoses with a perivascular vascular network—the remodelling of intramural channels. Cells located on the slopes of intramural canal entrances are often devoid of any orientation. There are small rounded ECs and giant ECs (Figure 2A–J).

Consequently, there is a fundamental possibility of the formation of new blood vessels directly from the endothelium of large blood vessels, in particular from the wall of the main vein. The development of intramural and endovasal structures increases with an increase in the degree of damage to the vascular wall. For example, when applying a microsurgical suture, most of them are located at the site of the joint. The proliferative activity of the newly formed endothelium remains in the mouths of intramural channels for a long time [84].

Similarly, the balloon-based injury caused an almost complete denudation of the intima of the cava veins in rats, and the lesions were fully covered with platelets 1 h after trauma. After 3 days, reendothelialisation had started. The fibrinolytic activity was significantly decreased at 1 and 24 h following trauma. Reendothelialisation was accompanied by return to normal fibrinolytic activity [87]. Interestingly, similar intramural angiogenesis is observed in lymphatic duct in vitro [88]. On the other hand, the perivenous application of PGE2 and glycerol induces the intense vascularisation of the wall of the rat femoral vein from its intimal ECs [89,90]. Venous ECs are enriched for the FUCCI-Negative state (early G1) and BMP signalling, while arterial endothelial cells are enriched for the FUCCI-Red state (late G1 phase) and TGF-β signalling [91].

## 4. Regeneration of the Lymphatic Endothelium

The ECs lining the thoracic lymphatic duct have a typical structure. In areas with laminar lymph flow, the BM of ECs does not contain pores. However, in zones of turbulent flow, the BM is purely developed [92,93,94,95,96,97]. In lymphatic vessels, the density of ECs is less than in arteries or veins. In contrast to arteries or veins in which the chromatin is segmented, in ECs of the lymphatic duct, the nuclear heterochromatin is granular and evenly distributed. Here, the BM of ECs does not have pores [92,94,98].

The cellular mechanisms of lymphatic regeneration are also rather unclear, because there are only a few works devoted to this aspect of cell and tissue biology [98,99,100]. In these papers, the regeneration of ECs of the thoracic duct has been experimentally tested in vivo. However, this issue is practically important, because recent improvements in microsurgical techniques on lymphatic vessels facilitated the treatment of lymphoedema. After the transplantation of skin graft to nude mice, donor lymphatic vessels appeared to spontaneously re-anastomose with recipient vessels. There was a spontaneous reconnection of recipient and donor lymphatic vessels. Then, lymphatic flow was restored [101].

When the canine and feline thoracic ducts were cryoinjured with the profiled 3 mm-based copper rod, the detachment of ECs damaged with ice crystals from basement membrane was significantly slower than in veins and aorta. The detachment of ECs occurs for 48 hrs after EC damage. On the surface of the BM, fibrin could be seen. Contacts between ECs became wider. The impregnation of the endothelium with AgNO_3_ revealed silver over these contacts. (Figure 3A,B). The ECs of neighbouring areas enter the S-phase and start their preparation to mitotic division (Figure 3C,D). Dividing ECs exhibited less evident protrusions of their bodies into the lumen of the duct and a lower development of microvilli and blebs. Lymphocytes were observed on the surface of endothelium and below ECs (Figure 3D,E). The important aspect of lymphatic EC regeneration is the mechanism of fibrin detachment from BM by the lamellipodia of ECs. Adjacent EC restored the defect within 3 days by migration and proliferation. We observed that on the first day, the endothelial monolayer included some elongated multinuclear cells with blind silver lines, whereas, on the third day, they were replaced by a population of smaller ECs with numerous mitoses. The organisation of the monolayer was restored within 7 days. The newly formed endothelium was similar to regenerating the endothelium of the arteries. In general, the clot that appeared at the zone of injury on the second day was dissolved by the third day. Occasionally, the dense polymorphic clot adhered to the wall and caused a delay in reendothelialisation [98,99,100].

The endothelial monolayer included some elongated multinuclear cells with blind silver lines whereas, on the third day, they were replaced by a population of smaller ECs with numerous mitoses. In three days, the restoration of the endothelial layer was completed. This occurs due to cell migration and proliferation at the margin of the cryoinjured zone. The organisation of the monolayer was restored within 7 days. The newly formed endothelium was similar to regenerating the endothelium of the arteries [102]. The molecular mechanisms involved into these events and related to arterial ECs, venous ECs and even lymphatic ECs in vitro are well-known [34,103,104]. Thus, cellular mechanisms involved into regeneration of ECs in different zones of the vascular bed are more or less the same. The main difference is related to the behaviour of the denuded area (Figure 4).

## 5. Comparison of EC Regenerations

Arterial, venous, and lymphatic vessels have common patterns of structural organisation and some difference in the orientation of ECs and the development of BM, namely, in arterial ECs, BM has pores. Vascular ECs restore small-size defects due to universal cellular mechanisms: the spreading of marginal ECs; the migration of intact cells into the damage zone; the proliferation of seemingly intact ECs situated near the injury. After damage, the ECs within the zone of injury undergo irreversible disorganisation due to defects in their PM. These defects are induced by drying, the formation of water crystals, toxins, cytosolic swelling, or mechanical pressure with different tools. The detachment of ECs induces the disappearance of tight junctions between the detached ECs and their live neighbours. The lamellipodia are formed by leading ECs. This induces the collective migration of ECs. Such spreading and subsequent EC migration can involve the attachment of ECs to the internal elastic membrane of BM if freezing or drying methods, or toxins, are used. ECs may start their division and undergo differentiation with a short stage of procollagen synthesis and transport for the generation of the BM. During this stage, procollagen-containing distensions inside ECs are described [38]. Presumably, this event would induce the overexpression of proteins forming intercellular junctions, namely, desmosomes, adherent junctions (such as different cell–cell adhesion proteins, like PECAM-1/CD, a calcium-dependent cell–cell adhesion glycoprotein VE-cadherins, and so on), and tight junctions (different occludins [ZO-1 and others]), claudins, junctional adhesion molecules, and angulins. The expression of angiopoietin receptors Tie1/2 and other stimulators of angio-genesis in leading ECs in tissues was not examined. It seems that the Notch signalling pathway is also involved. However, all these factors were studied mostly in vitro. It is not clear what molecular mechanisms are involved in the clearance of platelets from a vessel wall lacking ECs, or in the growth of ECs over the surface of thrombus.

This method, based on the freezing of the vessel wall, provided important information about the regeneration of vascular ECs in situ. Regeneration is realised by the movement of the EC directed towards the damage zone of the movement of the EC, originating from the edges of the viable endothelial layer. A sharp decrease in contact inhibition leads to the spreading of the edges of viable EC. This stimulates the second row of ECs to first enter the S-phase, then the G2 phase, and finally mitosis. In all three types of vessels, mitotically dividing ECs described using correlative light and electron microscopy have a body protruding into the lumen of the vessel, which is covered with microvilli and other outgrowths. The level of ECs is highest in the arteries and least pronounced in the lymphatic vessels. The intercellular contacts of mitotically dividing ECs become wider. The division of ECs leads to an increase in the density of these cells. They acquire a spindle shape. In the arteries, the endothelium, during regeneration, cleans the flattened platelets from the surface of the vessel, which are attached to the surface devoid of endothelium. In the veins, ECs grow on the surface of platelets and microthrombi.

The reactions of venous endothelial cells to damage differ little from those in arteries; however, the speed of response and the rate of reendothelialisation of venous vessels are significantly higher. In addition, the proliferating and migrating ECs are more spread out and do not have a clear spindle-like shape.

However, the lymphatic endothelium is involved in the repair process faster than the endothelium of veins and arteries. In lymphatic vessels, ECs are the slowest to split off from the basement membrane, and migrating ECs grow under fibrin filaments. After some time (usually after 30 days), the EC tissue mosaics return to normal. As a result, the closure of a defect in the wall of a lymphatic vessel occurs faster than in veins and 2–3 days faster than in arteries. The rate of reendothelialisation in lymphatic vessels and veins is affected by the low rate of lymph flow and the absence of adhered platelets on the surface of the damage zone, which form a thrombus in the arteries and veins. After damage, a fibrin thrombus forms in the lymphatic vessels, which persists until the defect is completely reendothelialised. The whole regeneration process takes place under it. Fibrin above the regeneration area is able to carry out a protective function, limiting the focus of inflammation, attracting phagocytes, and stimulating the synthetic activity of endothelial cells. Ageing, hypertension, and other pathological conditions of veins and lymphatic vessels have not been thoroughly investigated. It is not clear how small arteries and veins regenerate after intraluminal injections. Capillaries, on the other hand, do not regenerate but undergo angiogenesis [5].

## 6. Conclusions

The regeneration of ECs lining arteries, veins, and large lymphatic vessels is important in pathology. A comparison of these processes reveals both similar features and certain differences, most of which are related to the mode of interaction between migrating ECs and the vascular wall lacking ECs. Unfortunately, the molecular mechanisms explaining these differences are unknown. These issues should be examined.

## Figures and Tables

**Figure 1 ijms-26-00837-f001:**
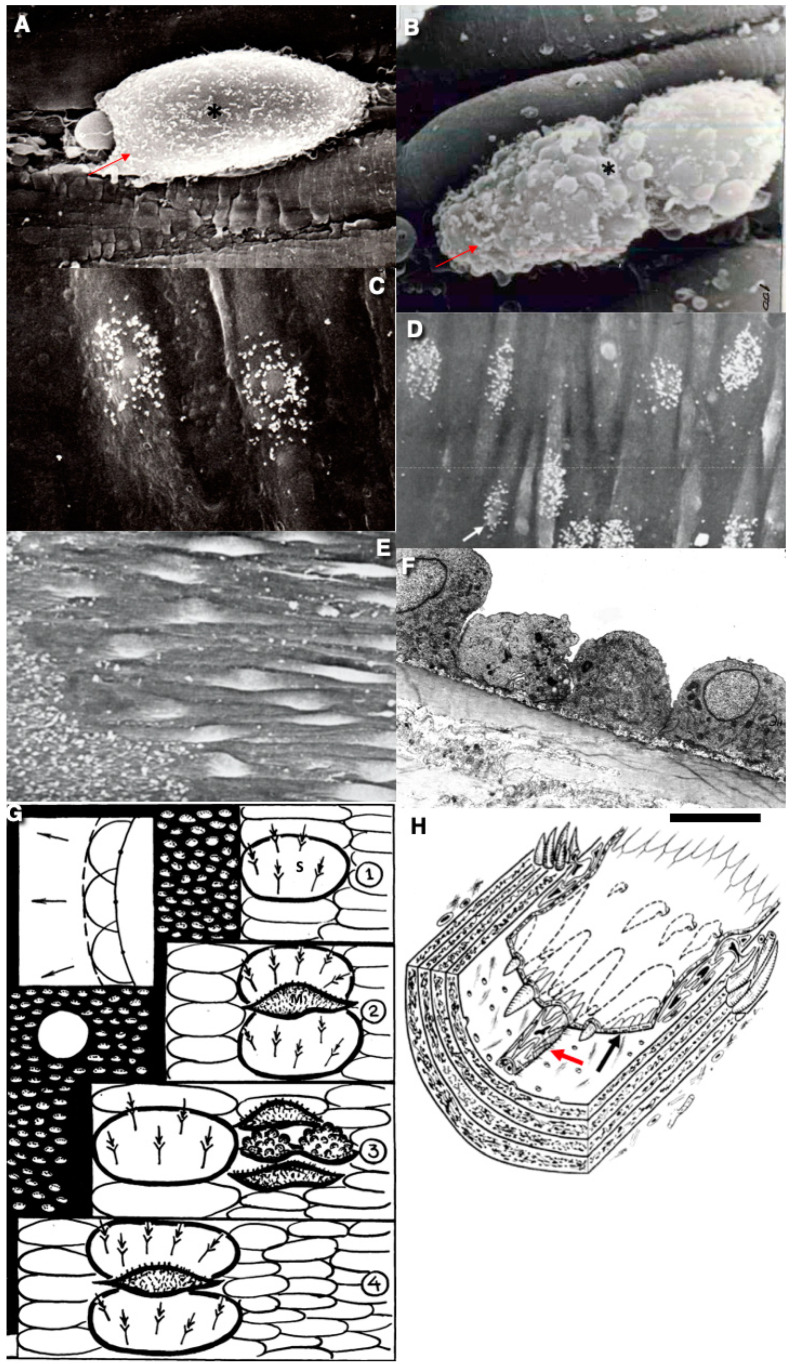
The features of the regeneration of ECs. (**A**,**B**) mitotic ECs: (**A**) ECs in prophase; (**B**) ECs in telophase. Scanning electron microscopy (SEM). Dividing ECs are shown with red arrows and asterisks. (**C**,**D**) Autoradiography on the basis SEM method demonstrates one of ECs being in S-phase (white arrow). Silver granules (white dots) show the localisation of the H^3^ isotope. (**E**) Spindle-shaped ECs in the zone of the endothelial regeneration. (**F**) A cross-section of the spindle-shaped ECs situated within the regeneration zone. (**G**) The scheme of the EC dynamics according to the wave-like process. Numbers 1–4 indicate stages of the process. (**H**) A scheme of the regeneration of SMCs in the aorta after extensive cryotherapy. SMCs form fusiform structures (red arrow) moving in the space below ECs. The black arrow points to the endothelial layer. SMCs were not observed in the medial layer because of their destruction by water crystals. The images (**A**–**E**) are taken from Figure 1, presented in [38]. The image (**F**) is taken from Figure 2K, presented in [38]. The drawings are reprinted under a Creative Commons licence (the copyright licence is a non–commercial publication). Band size: 5 microns (**A**–**F**).

**Figure 2 ijms-26-00837-f002:**
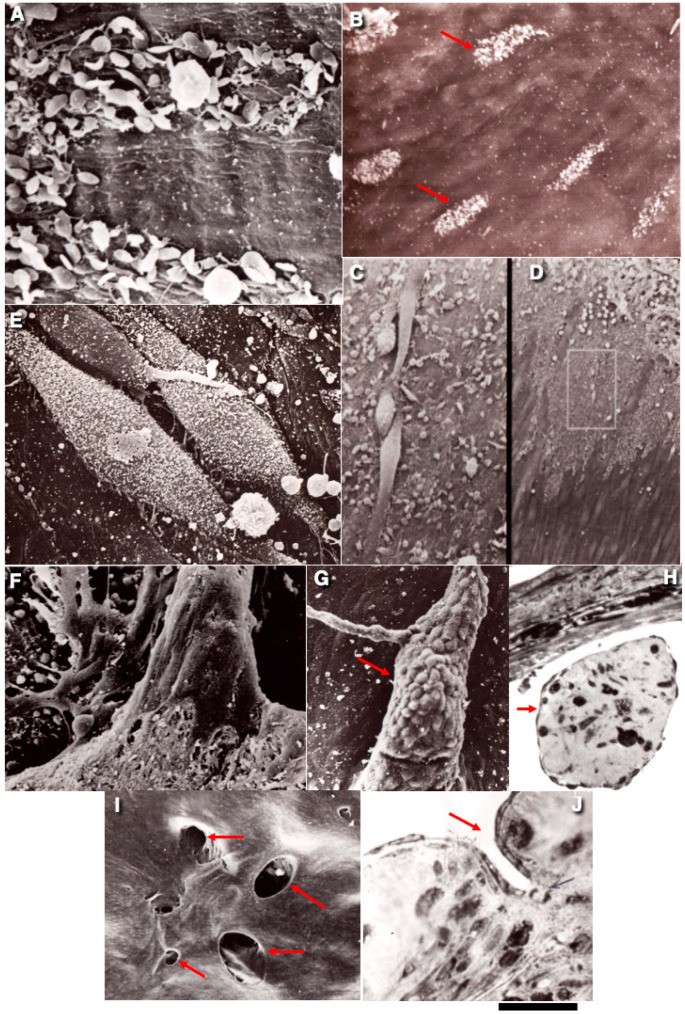
The features of ECs during the regeneration of endothelium in vena cava inferior. (**A**) Leading ECs during regeneration of endothelium. ECs move on the surface of activated platelets. (**B**) SEM autographic labelling (white dots and red arrows) of ECs in S-phase. (**C**,**D**) Isolated ECs situated ahead of the collective EC layer. The image in (**D**) is an enlargement of the white box in (**E**). (**E**) ECs in mitosis. In the inferior vena cava, mitotic ECs are more flattened than in aorta. This is caused by the fact that BM is solid, whereas in the aorta BM appears as a mesh. (**F**,**G**) “Endovasal” structure (red arrow) covered with ECs. (**H**) A cross-section of the “endovasal” structure (red arrow). (**I**) Intramural angiogenesis into the vein wall after the mechanical damage of vena cava inferior. Entrances (red arrows) into angiogenic channels are visible. (**J**) A cross-section of the channel near its entrance (red arrow). The images in (**A**–**J**) are taken from [83,84]. The figures are reprinted courtesy of Creative Commons License (Attribution–Noncommercial–Share Alike 4.0 Unported licence). Bars: 5 µm (**A**–**G**): 25 µm (**H**–**J**).

**Figure 3 ijms-26-00837-f003:**
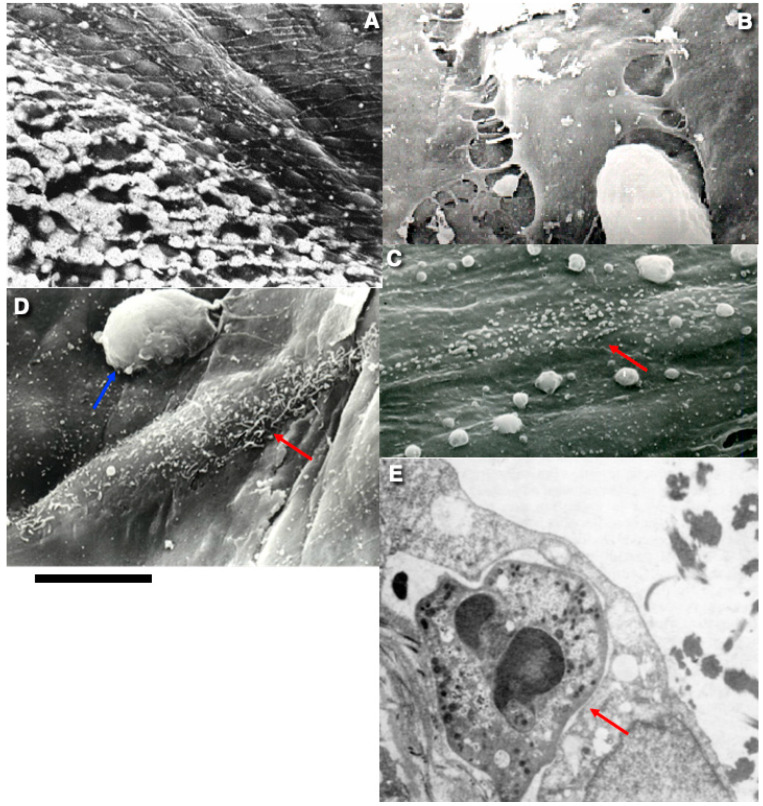
The regeneration of lymphatic ECs. (**A**) After the freezing of the thoracic duct, ECs acquired the ability to be stained with AgNO3. The impregnation of ECs with AgNO3. (**B**) Contacts between ECs became wider. (**C**,**D**) In the thoracic lymphatic duct, mitotic ECs (red arrows) are more flattened than in an aorta and protrude less into the lumen (see Figure 1A,B) because, in the duct, BM is solid, whereas in an aorta BM forms a mesh with pores. In addition, in the thoracic duct, ECs are tightly bound to the collagen fibrils within BM with sling/strap filaments. In (**D**), a lymphocyte (blue arrow) is visible in the middle of the upper part of the image. In (**E**), a lymphocyte (red arrow) is visible below EC. Images (**A**–**E**) are taken from [98,99,100,102]. The figures are reprinted courtesy of Creative Commons License (Attribution–Noncommercial–Share Alike 4.0 Unported licence). Bars: 15 µm (**A**–**D**); 5 µm (**E**).

**Figure 4 ijms-26-00837-f004:**
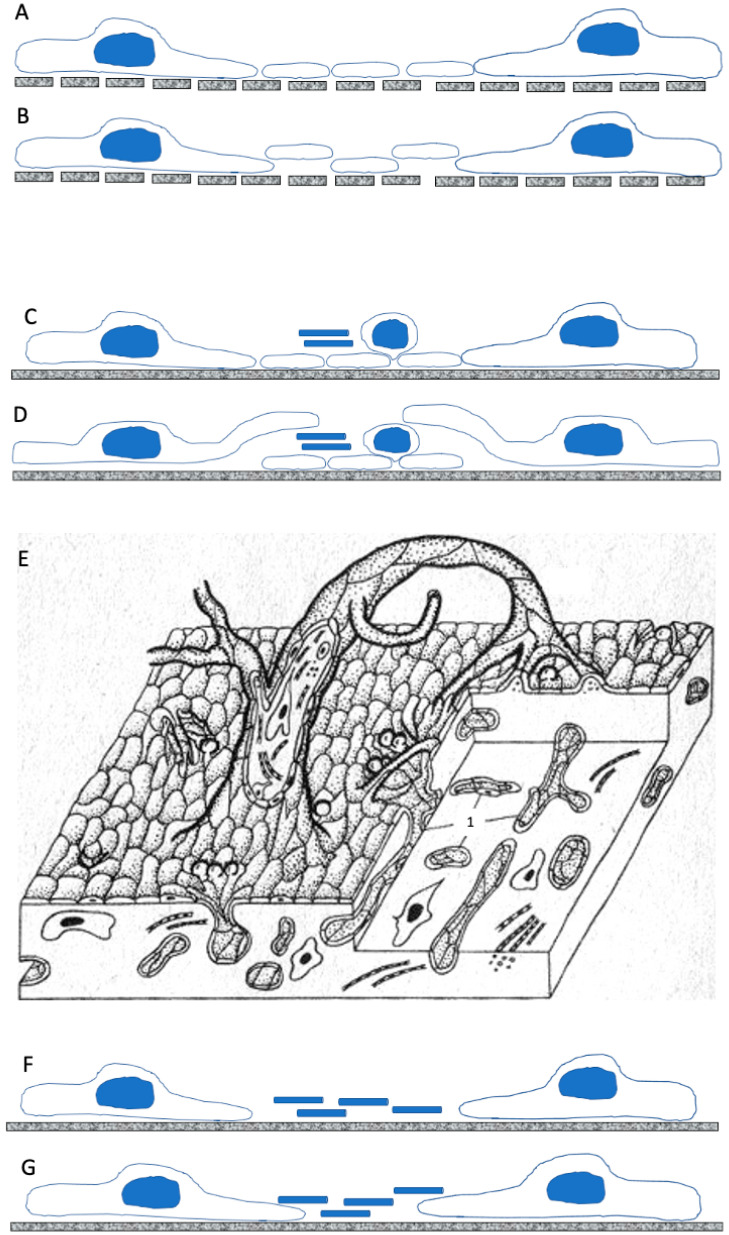
The schemes of endothelial regeneration in the aorta (**A**,**B**), vein (**C**–**E**) and lymphatic duct (**F**,**G**). After the denudation of ECs, the luminal surface of the aorta is covered with platelets (in the middle). Healthy ECs surround platelets. (**B**) During regeneration, the lamellipodia of ECs eliminate platelets from mesh-like BM. (**C**) After the denudation of ECs damaged by water crystals, the luminal surface of vena cava inferior is covered with platelets, leukocytes, and fibrins (blue cylinders). (**D**) ECs move over the layer of platelets and fibrin bundles. (**E**) A scheme of the microthrombus over the injured area. Fibrin bundles are inside this “endovasal” structure. Number 1 indicates the section of intramural pseudo-vessels. (**F**) After the freezing of the wall and EC denudation, the surface of the lymphatic duct is covered with fibrin. (**G**) ECs induce the detachment of fibrin bundles from the surface of the duct. Image (**E**) is taken from references [55,68].

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
