# Peer review of "Regeneration of Vascular Endothelium in Different Large Vessels"

_ijms, 2025, doi:10.3390/ijms26020837_

Round 1
Reviewer 1 Report
Comments and Suggestions for Authors
Many thanks for giving me the possibility to review the valuable manuscript that aims to review mechanisms of endothelial regeneration in arterial, venous and lymphatic vessels.
It is my opinion that the structure of the manuscript needs to be reorganized for a better readability.
At first, the purpose of the study needs be better disclosed.
The amplitude of the topic requires that the contents be expressed in a concise and clear manner otherwise the reader risks being confused at the end of reading.
Usually conclusion does not contain figures and references. It is better to move them in discussion paragraph.
Figures need a better description. For example in figure 1, H is not described. Instead in figure 2 K description is not linked to any figure.
Conclusions seem not consistent with presented data and add no new data over current literature. They seem a repetition of what previously discussed.
Comments on the Quality of English LanguageThere are several orthographic errors, repetitions and some non-sense sentences. The manuscript needs a thorough revision to improve the English language quality.
Author Response
Editor
I have noted that your paper does not contain a "Math & Meth" section. Furthermore, the format does not correspond to the author instructions asked for by the journal.
"Article: original research manuscripts. The work should report scientifically sound experiments and provide a substantial amount of new information. The article should include the most recent and relevant references in the field. The structure should include an Abstract, Keywords, Introduction, Materials and Methods, Results, Discussion, and Conclusions (optional) sections." We need these format revisions before considering your paper for peer-review.
Reply
Thanks a lot for your comments. We changed the format of this article from an article into a review.
Reviewer 1
Many thanks for giving me the possibility to review the valuable manuscript that aims to review mechanisms of endothelial regeneration in arterial, venous and lymphatic vessels.
- It is my opinion that the structure of the manuscript needs to be reorganized for a better readability.
Reply
We tried to improve this structure.
- At first, the purpose of the study needs to be better disclosed.
Reply
We proposed a new purpose.
- The amplitude of the topic requires that the contents be expressed in a concise and clear manner otherwise the reader risks being confused at the end of reading.
Reply
We made the paper shorter and more concise and clearer.
- Usually, conclusion does not contain figures and references. It is better to move them in discussion paragraph.
Reply
We corrected "Conclusion", namely, eliminated references and the Figure.
- Figures need a better description. For example, in figure 1, H is not described. Instead in figure 2 K description is not linked to any figure.
Reply
We re-wrote legends and corrected mistakes.
- Conclusions seem not consistent with presented data and add no new data over current literature. They seem a repetition of what previously discussed.
Reply
We eliminated repetitions. The aim of the review is to remind old important although well-forgotten data and compare cell mechanisms of arterial, venous and lymphatic EC regeneration
- There are several orthographic errors, repetitions and some non-sense sentences. The manuscript needs a thorough revision to improve the English language quality.
Reply
We corrected the text.

Reviewer 2 Report
Comments and Suggestions for Authors
The manuscript by Sesorova et al reviews mostly clinically related aspects of endothelial regeneration in big arteries, veins and lymphatics. I found this manuscipt potentially interesting and useful for the audience of the Journal.
I'm personally more interested in molecular mechanisms that can drive or affect re-endothelization. Therefore I would ask the authors provide a brief overview about endothelial junctional proteins during re-endothelization, such as VE-cadherins, claudins, Pecam1, Dll/Notch, Tie1/2. Is their expression increased or decreased?
Also, some brief overview on the vascular growth factors involved (VEGF, VEGF-C, PlGF)?
Other cytokines involved?
When comparing curved areas of the aorta (aortic arch, which is atheroprone) to the linear (atheroresistant) parts, does the aortic shape affect regeneration?
In practical conditions, such as ischemia-reperfusion, where hypoxia is induced (stimulating Hif expression), does this condition promote regeneration?
Various notes for the text:
Line 65: Should be 350-µm (not 350-m)
Line 81: "PCs" needs to be spelled-out.
Line 88: double "that"
Line 93: the word "remind" would be more correct than "remain"
Line 98: referenced work by Cotton et al should be 4, not 4/4
Line 126: instead of "are" should be "area"
Do the authors use "endothelial cells" and "endotheliocytes" as synonimous?
Line 184: perhaps some word is missing?
Lines 188-189 are the repetition of lines 135-136.
Line 191: suitable reference(s) would be needed.
Lines 192-194: what kind of "tumour-derived factors" are meant?
Line 197: this part of the sentence need some improvement (perhaps, some word is missing)
Line 198: "GC" needs to be spelled-out.
Lines 168-171 are the direct repetition of lines 206-208.
Line 203: LDL is not spelled-out, whereas later (line 222) it is spelled-out.
Line 227: what does reference 54/34 mean?
Line 269: 55/35 meaning?
Line 377: "damages" should be replaced with "damaged"
Line 378: "of" should be removed
Author Response
Reviewer 2
The manuscript by Sesorova et al reviews mostly clinically related aspects of endothelial regeneration in big arteries, veins and lymphatics. I found this manuscript potentially interesting and useful for the audience of the Journal.
Reply
Thanks a lot for your job.
- I'm personally more interested in molecular mechanisms that can drive or affect re-endothelization. Therefore, I would ask the authors provide a brief overview about endothelial junctional proteins during re-endothelization, such as VE-cadherins, claudins, Pecam1, Dll/Notch, Tie1/2. Is their expression increased or decreased?
Reply
We included a very brief part on these mechanisms. Unfortunately, most data were obtained in vitro and were nor checked in tissue situation. Although all these factors are described in vitro in tissue almost nothing is known.
- Also, some brief overview on the vascular growth factors involved (VEGF, VEGF-C, PlGF)?
Other cytokines involved?
Reply
These factors are involved but data are not directly obtained from in tissue situation.
- When comparing curved areas of the aorta (aortic arch, which is atheroprone) to the linear (athero-resistant) parts, does the aortic shape affect regeneration?
Reply
We included these data. The shape is important because it induces different modes of blood flow. Also, the pressure and its pulsations are important.
- In practical conditions, such as ischemia-reperfusion, where hypoxia is induced (stimulating Hif expression), does this condition promote regeneration?
Reply
We included data obtained in our laboratory and describing the effect of hypoxia.
- Various notes for the text:
Line 65: Should be 350-µm (not 350-m)
Line 81: "PCs" needs to be spelled-out.
Line 88: double "that"
Line 93: the word "remind" would be more correct than "remain"
Line 98: referenced work by Cotton et al should be 4, not 4/4
Line 126: instead of "are" should be "area"
Reply
We eliminated all these mistakes and unclearness
- Do the authors use "endothelial cells" and "endotheliocytes" as synonymous?
Reply
We eliminated the word endotheliocytes
7.
Line 184: perhaps some word is missing?
Lines 188-189 are the repetition of lines 135-136.
Line 191: suitable reference(s) would be needed.
Lines 192-194: what kind of "tumour-derived factors" are meant?
Line 197: this part of the sentence need some improvement (perhaps, some word is missing)
Line 198: "GC" needs to be spelled-out.
Lines 168-171 are the direct repetition of lines 206-208.
Line 203: LDL is not spelled-out, whereas later (line 222) it is spelled-out.
Line 227: what does reference 54/34 mean?
Line 269: 55/35 meaning?
Line 377: "damages" should be replaced with "damaged"
Line 378: "of" should be removed
Reply
We eliminated all these mistakes and unclearness.
